# Household survey on prevalence and risk factors for obesity in owned cats from Central Brazil

**Danilo Conrado Silva**[1,2*], **Mariana Yukari Hayasaki Porsani**[2], **Aparecido Divino da Cruz**[3], **Erika Figueiredo Pereira**[4], **Klayto José Gonçalves dos Santos**[5], **Lysa Bernardes Minasi**[3], **Vitória Alvarenga Nunes**[3], **Alex Silva da Cruz**[3], **Fabio Alves Teixeira**[2,4]

1 Veterinary Medicine, West Campus, State University of Goiás, São Luís de Montes Belos, Goiás, Brazil, 2 National Association of São Paulo for Small Animal Practice (ANCLIVEPA-SP), São Paulo, Brazil, 3 Replicon Research Group, Genetics Graduate Program, School of Medical and Life Sciences, Pontifical Catholic University of Goiás, Goiânia, Goiás, Brazil, 4 School of Veterinary Medicine and Animal Science, University of São Paulo, São Paulo, Brazil, 5 Center for Animal Reproduction Biotechnology (Biotec), Graduate Program in Animal Production and Forage Science, West Campus, State University of Goiás, São Luís de Montes Belos, Goiás, Brazil

* danilo.silva@ueg.br

## Abstract

Few studies have conducted the prevalence of obesity in cat populations and the associated risk factors by assessing cats in their homes, regardless of whether they had visited a veterinary hospital. Moreover, such studies are scarce in Latin America, and, to date, few census-based, in-home epidemiological assessment of feline obesity has been conducted in Brazil. This study aimed to estimate the prevalence of obesity in owned cats in the metropolis of Goiânia, Goiás, Brazil, and to assess the presence of risk factors related to the animals, their owners, the home environment, and cat management practices. A cross-sectional study was conducted with 188 cats and their owners, using geographically stratified household sampling. The animals were categorized based on body condition score (BCS) as underweight, ideal weight, overweight, or obese, and prevalence rates were calculated. The BCS was investigated for its association with animal, owner, environmental, and management characteristics using the Kruskal-Wallis test ($\alpha = 5\%$). Subsequently, characteristics with $p \leq 0.05$ were analyzed as potential risk factors for overweight or obesity using binary logistic regression. Kappa analysis was used to determine the agreement between the BCS determined by veterinarians and the owners' perception of their cats' body condition. The prevalence of overweight and obesity in cats was 28.7%. Factors related to age and activity level, type of household, as well as the owners' education and occupation, were identified as risk factors for the development of overweight or obesity in the study cat population. Additionally, protective factors against overweight and obesity were identified, including the number of cats in the household, the location of the household, and the owners' perception of their own diet. The agreement

**Data availability statement:** All relevant data are within the manuscript.

**Funding:** The authors wish to thank ROYAL CANIN DO BRASIL for their financial contribution toward the article processing charge (APC). This funding was sought and secured only after the manuscript had been accepted for publication; therefore, the sponsor had no influence on the research process, data analysis, or editorial content of this publication

**Competing interests:** The authors wish to thank ROYAL CANIN DO BRASIL for their financial contribution toward the article processing charge (APC). This does not alter our adherence to PLOS ONE policies on sharing data and materials.

between the actual BCS of the animals and the owners' perception of their cats' body condition was considered low.

## Introduction

Obesity is the most common nutritional disorder in small animal veterinary clinical practice [1,2]. Excessive food intake causes the accumulation of excessive body fat, leading to numerous health problems and reduced life expectancy. Obesity in cats is associated with genetic, environmental, behavioral, sociocultural factors, and even factors directly related to their owners, such as socioeconomic status, eating habits, age and body condition [1–3].

Obesity is a multifactorial disease, although it is closely related to diet and feeding management that lead to a positive energy balance [4]. In veterinary medicine, risk factors for obesity inherent to animals are also described, such as breed, age, sex, reproductive status, hormonal influences, and low physical activity [2,5].

In the United Kingdom, United States, New Zealand, Australia, France, and the Netherlands, the prevalence of overweight and obesity in cats has been estimated to range from 11.5% to 63% [6–15], showing a large variation due to diverse population conditions, as well as methodological and sampling variations. These studies identified potential risk factors for the development of obesity in cats, including breed, age, sex, reproductive status, relationship with the owner, owners' perception of their cats' body condition, diet type, feeding frequency, and environment.

Few publications on the prevalence of feline obesity in Brazil were identified, but their findings are discrepant. A study conducted in a small town (Alegre, Espírito Santo), which involved household visits, reported a prevalence of overweight cats at 14%, with 6% classified as obese [16]. In contrast, a study carried out in a major metropolitan area, in the city of Rio de Janeiro, found that nearly 60% of cats presenting to a veterinary hospital for surgical procedures were overweight, with 36.8% classified as obese [17]. Finally, in the southern region of the country (Porto Alegre, Rio Grande do Sul), another study evaluating cats seen at a veterinary hospital—specifically during the COVID-19 pandemic— found yet another different value: 38.4% of cats were overweight, with only 8.7% classified as obese [18].

The reported risk factors for feline obesity were largely consistent across both studies [16,17]. Middle-aged, neutered male cats, particularly those living in households with free access to food, were more likely to be obese. Additionally, owner underestimation of the cat's body condition score (BCS) contributed as a predisposing factor [17]. In both studies, male sex and neutering were associated with increased obesity risk, whereas physical activity was identified as a protective factor [16].

Despite prevalence studies and risk factors related to obesity in cats having been conducted in different locations around the world, data are scarce in Latin America, and there is limited data from Brazil, especially using a census-based approach with

home visits. Therefore, the aim of this study was to estimate the prevalence of obesity in owned cats in the city of Goiânia, state of Goiás, Brazil, and to examine risk factors related to the animals, their owners, the home environment, and cat management practices.

## Materials and methods

### Sample group and ethical considerations

A cross-sectional study was conducted in the city of Goiânia, Brazil, with owned cats as the experimental units to determining the prevalence of obesity and associated risk factors. In addition, data were collected from the animals and their owners, as well as from management practices. The study was conducted between March and September of 2023. This study was approved by the Animal Ethics Committee (approval number 014/2022) and the Research Ethics Committee (approval number 5.925.047) of the State University of Goiás. Data collection was performed after the pet owners included in the study signed the informed consent form.

The minimum number of animals required to determine the prevalence of obesity and associated risk factors was defined using the sample size equation (n) based on the expected prevalence (EP), where $n = EP \times (1 - EP) \times (1.96)^2 / 0.05^2$. An EP of 14% for obesity in cats was considered, based on the only study that estimated the prevalence of owned cats through home visits sampling in Brazil [6]. A margin of error of 5% and a 95% confidence level were used. Therefore, the minimum sample size was calculated, resulting in 185.01, thus minimum n = 186 cats. Cats under 1 year of age were excluded from the study due to the focus on assessing the prevalence of obesity in adult cats.

Sampling continued until two of the following criteria were met. The first criterion was a minimum of 186 cats. The second criterion was household sampling through geographic stratification, ensuring that the proportion of households in each weighting area of the city of Goiânia was approximately maintained [19]: South (19.7%), Central (18.8%), Southwest (16.4%), East (12.1%), Northwest (11.3%), West (11.2%), and North (10.5%).

Before sampling in each weighting area, neighborhoods and streets to be visited were randomly selected. The first household visited in each street was always the one with the lowest number, continuing to the immediately adjacent house if sampling could not be conducted for any reason. In households with more than one cat, all cats were evaluated. In cases where two or more owners were present, only the one most involved in the animals' management was assessed.

### Data collection and categorization of cats and owners

Each selected owner completed a questionnaire to collect personal and family data, as well as information about the cats and their management (Box 1). Owner data included eating habits, physical activity, and socioeconomic status. Owners' eating habits were classified according to the Dietary Guidelines for the Brazilian Population [20], with "unhealthy" eating habits defined as consuming snacks three or more times per week or consuming fruits and vegetables two or fewer times per week. For the purpose of stratifying owners by household income, amounts in the Brazilian currency, the Brazilian real, were converted into US dollars.

**Box 1.    Questionnaire answered by owners included in the study**

| 1 – Your age | 2 – Gender | 4 – Education |
|---|---|---|
| a) 18–24 years | a) Male | a) Elementary School |
| b) 25–34 years | b) Female | b) Middle School |
| c) 35–44 years | 3 – The house where you live is? | c) High School |
| d) 45–59 years | a) Owned | d) Collage/University |
| e) 60–75 years | b) Rented | e) Specialization |
| f) Older than 76 years | c) Tenant without a lease | f) Did not attend school |

| 5 – Total monthly household income | 6 – Employment Status | 7 – How many weekly hours do you work? |
|---|---|---|
| a) No income<br>b) Up to $ 225<br>c) From $ 225 to $ 670<br>d) From $ 670 to $ 1.790<br>e) More than $ 1.790 | a) Government (Public Sector)<br>b) Company (Private or State)<br>c) Non-governmental organization<br>d) Self-employed<br>e) Rural Property<br>f) Unemployed<br>g) Retired<br>h) Homemaker<br>i) Student | a) No set work hours, up to 10 hours per week<br>b) From 11 to 30 hours per week<br>c) From 30 to 40 hours per week<br>d) More than 40 hours per week |
| 8 – Number of people that live with you and their age (including employees)<br>a) Between 0 and 12 years<br>b) Between 13 and 18 years<br>c) Between 19 and 24 years<br>d) Between 25 and 34 years<br>e) Between 35 and 44 years<br>f) Between 45 and 59 years<br>g) Between 60 and 75 years<br>h) Older than 76 years | 9 – Your physical activity frequency<br>a) Daily<br>b) Once a week<br>c) Three times a week<br>d) More than three times a week<br>e) Does not exercise | 10 – Do you believe you eat<br>a) In excess<br>b) Normally<br>c) Low amounts<br>**11 – How many times do you consume fried food?**<br>a) Daily<br>b) 1–2 times a week<br>c) 3 or more times a week<br>d) Occasionally<br>e) Never |
| 12 – How many times do you consume fruits?<br>a) Daily<br>b) 1–2 times a week<br>c) 3 or more times a week<br>d) Occasionally<br>e) Never | 13 – How many times do you consume vegetables?<br>a) Daily<br>b) 1–2 times a week<br>c) 3 or more times a week<br>d) Occasionally<br>e) Never | 14 – How many times do you consume snacks?<br>a) Daily<br>b) 1–2 times a week<br>c) 3 or more times a week<br>d) Occasionally<br>e) Never |

https://doi.org/10.1371/journal.pone.0337397.t0010

Data on the animals included basic information such as age, sex, breed, and reproductive status, as well as sanitary and nutritional management practices. Additionally, the owner's perception of their cats' current body condition was also collected. Owners chose their cats' body condition by selecting one of the following categories in the questionnaire: underweight, ideal weight, overweight, or obese.

The cats were evaluated by veterinarians using the nine-point BCS [21] and classified as underweight (BCS 1–4), ideal weight (BCS 5), overweight (BCS 6–7), and obese (BCS 8–9). Age classification followed the Feline Life Stage Guidelines [22]: 1–6 years – young adult; 7–10 years – mature adult; over 10 years – senior.

Owner's body mass index (BMI) was calculated using the methodology recommended by the World Health Organization [23] through the equation: BMI = weight (kg)/ height² (m²). Height was measured with a tape measure, and weight was measured on a portable digital scale. BMI was classified as underweight (<18.5), normal weight (18.6–24.9), overweight (25.0–29.9), and obesity (≥30.0). Anthropometric measurements were also taken using a tape measure, including abdominal, waist and hip circumferences. A waist/height ratio of >0.52 was considered a potential health risk, while waist/hip ratios of <0.91 for men and <0.76 for women were considered low risk for cardiovascular disease [24].

## Statistical analysis

The results were presented and analyzed through frequencies distributed among the determined cats' BCS (underweight, ideal weight, overweight, and obesity), as well as the characteristics of the animals (intrinsic and sanitary characteristics), the owners (intrinsic, anthropometric, lifestyle habits, socioeconomic characteristics, and perception of the animals' body condition), and the cats' dietary management.

To assess potential differences in the distribution of BCS frequencies across each characteristic analyzed, the Kruskal-Wallis test was performed with a significance level of 5%. Subsequently, characteristics with p ≤ 0.05 were analyzed as

potential risk factors for being overweight or obesity using binary logistic regression (BLR), from which odds ratios (OR) and 95% confidence intervals were calculated.

Kappa analysis was used to determine agreement between BCS determined by veterinarians and the owners' perception of their cats' body condition. The agreement is considered low if between 0.00 and 0.20; reasonable if between 0.21 and 0.40; moderate if between 0.41 and 0.60; high if between 0.61 and 0.80; and almost perfect if between 0.81 and 1.00 [1].

All statistical analyses were conducted using statistical software (SPSS, version 20; IBM Corporation).

## Results and discussion

This study included 188 cats from 80 households, with an average of 2.35 cats per household. A total of 1043 households were visited, but 963 were excluded due to various reasons: no residents present at the time of the visit (547), households without cats (318), aggressive or missing cats (71), and owners who decline to participate (27).

The 80 households included in the study were sampled across the seven weighting areas of Goiânia, distributed as follows: 16 in the South (20%), 15 in the Central (18.75%), 13 in the Southwest (16.25%), 10 in the East (12.5%), 9 in the Northwest (11.25%), 9 in the West (11.25%), and 8 in the North (10%).

The BCS assessment classified 57 cats (30.3%) as underweight, 77 (41%) as ideal weight, 38 (20.2%) as overweight, and 16 (8.5%) as obese. Thus, the total prevalence of overweight and obese cats was 28.7%, with 20.2% classified as overweight and 8.5% as obese. Studies from various countries have estimated the prevalence of obesity in cats across different populations, with overweight and obesity rates ranging from 11.5% to 63% [6–15]. According to Tarkosova et al. [25], this wide variation in prevalence can be attributed to the specific characteristics of each studied population such as geographic region, differences in feeding practices, and the proportion of neutered or indoor cats.

Our values are intermediate compared to the prevalence rates of overweight and obesity reported in other Brazilian studies, which ranged from 14% to 38.4% [16–18]. They are more similar to the only study that also conducted home visits, which found 6% of cats to be obese [16]. Despite the similarity, this study was carried out in a city with a human population approximately 50 times smaller than that of Goiânia [19].

The distribution of BCS categories was compared with the intrinsic characteristics of the cats in the study (Table 1). Differences (p < 0.05) in distribution of BCS frequencies across each characteristic were observed based on age group, reproductive status, age at neutering, and the level of physical activity reported by the owner.

In contrast to previous studies that identified association between obesity and male cats [8,10,11,15–17], it was not found in the present study, despite the higher frequency of obese males (Table 1). Neutering has been described in the literature as an important risk factor for overweight and obesity in cats [7,8,10,11,13,17,26]. In the present study, despite the reproductive status statistically influencing the cats' BCS (Table 1) and reproductive status seems to influence the cats' BCS due to the higher frequency of intact cats with a BCS < 4, neutering was not a risk factor for overweight or obese cats (p = 0.161) (Table 2). A recent study [26] discusses the relationship between the age at neutering and its effect on feline body condition. In our data, the distribution of body condition score (BCS) also differed according to the age at neutering (Table 1).

Middle-aged cats, between 7 and 10 years old, had the highest frequency of overweight and obesity (Table 1) and OR = 3.5 (Table 2), supporting the literature [9–11,14,17,27], that showed the prevalence of obesity in cats tends to increase up to 10 years of age and then decreases.

The activity levels of the cats reported by their owners influenced the cats' BCS (Table 1). Greater inactivity was shown to be a risk factor for overweight and obesity (OR = 2.7; Table 2). Increased opportunities for physical activity have already been described as a protective factor against obesity in cats [16,28]. In a cross-sectional study conducted in New Zealand [12], cats were categorized as inactive, normal, or hyperactive, with the prevalence of overweight/obesity found to be 81%, 63%, and 46%, respectively (p = 0.037).

**Table 1. Distribution of body condition score (BCS) frequencies accordingly with each intrinsic characteristic of owned cats in Goiânia, Brazil.**

| Intrinsic Characteristic | Underweight (BCS 1–4) | | Ideal (BCS 5) | | Overweight (BCS 6–7) | | Obese (BCS 8–9) | | Total | | p-value* |
|---|---|---|---|---|---|---|---|---|---|---|---|
| | N | (%) | N | (%) | N | (%) | N | (%) | N | (%) | |
| **Sex** | | | | | | | | | | | |
| Male | 25 | (32.1) | 34 | (43.6) | 10 | (12.8) | 9 | (11.5) | 78 | (41.5) | 0.147 |
| Female | 32 | (29.1) | 43 | (39.1) | 28 | (25.4) | 7 | (6.4) | 110 | (58.5) | |
| **Age range** | | | | | | | | | | | |
| 1 to 6 years | 37 | (27.2) | 64 | (47.1) | 28 | (20.6) | 7 | (5.1) | 136 | (74.3) | 0.009 |
| 7 to 10 years | 12 | (34.3) | 8 | (22.9) | 8 | (22.9) | 7 | (20.0) | 35 | (19.1) | |
| More than 10 years | 4 | (33.3) | 4 | (33.3) | 2 | (16.7) | 2 | (16.7) | 12 | (6.6) | |
| **Reproductive status** | | | | | | | | | | | |
| Neutered | 36 | (24.3) | 63 | (42.6) | 33 | (22.3) | 16 | (10.8) | 148 | (78.7) | 0.002 |
| Intact | 21 | (52.5) | 14 | (35.0) | 5 | (12.5) | 0 | (0.0) | 40 | (21.3) | |
| **Sex and reproductive status** | | | | | | | | | | | |
| Neutered male | 14 | (25.0) | 25 | (44.6) | 8 | (14.3) | 9 | (16.1) | 56 | (29.8) | 0.084 |
| Intact male | 11 | (50.0) | 9 | (40.9) | 2 | (9.1) | 0 | (0.0) | 22 | (11.7) | |
| Neutered female | 22 | (23.9) | 38 | (41.3) | 25 | (27.2) | 7 | (7.6) | 92 | (48.9) | |
| Intact female | 10 | (55.5) | 5 | (27.8) | 3 | (16.7) | 0 | (0.0) | 18 | (9.6) | |
| **Age at neutering** | | | | | | | | | | | |
| Up to 1 year | 5 | (27.8) | 3 | (16.7) | 6 | (33.3) | 4 | (22.2) | 18 | (27.3) | 0.005 |
| More than 1 year | 3 | (37.5) | 1 | (12.5) | 3 | (37.5) | 1 | (12.5) | 8 | (12.1) | |
| Intact cats | 24 | (52.5) | 16 | (35.0) | 6 | (12.5) | 1 | (0.0) | 47 | (60.6) | |
| **Breed** | | | | | | | | | | | |
| Mixed breed | 57 | (31.5) | 74 | (40.9) | 36 | (19.9) | 14 | (7.7) | 181 | (96.3) | 0.118 |
| Purebred | 0 | (0.0) | 3 | (43.0) | 2 | (28.5) | 2 | (28.5) | 7 | (3.7) | |
| **Physical activity** | | | | | | | | | | | |
| Active | 38 | (31.9) | 55 | (46.2) | 20 | (16.8) | 6 | (5.1) | 119 | (63.3) | 0.031 |
| Inactive | 19 | (27.5) | 22 | (31.9) | 18 | (26.1) | 10 | (14.5) | 69 | (36.7) | |

BCS: Body Condition Score; *p value obtained by the Kruskal-Wallis test.

The breed was not statistically associated with BCS (Table 2), which may be explained by the small sample size of purebreds cats. According to Courcier et al. [11], breed was not considered a risk factor for overweight or obesity in cats in Great Britain. However, other studies have shown a higher susceptibility in certain breeds as Norwegian Forest Cat, British Shorthair and Persian [9,13] or in cats with no defined breed [8,15,27].

In Brazil, the frequency of overweight and obesity was also assessed in two samples populations of cats visiting veterinary hospitals. A population of 106 cats undergoing surgical procedures at a feline private veterinary clinic in Rio de Janeiro [17] and a population of 389 cats visiting a teaching hospital in Porto Alegre [18]. Among the evaluated cats, 60.4% and 38.4% were above ideal weight, with 23.6% and 29.7% classified as overweight, and 36.8% and 8.7% as obese, respectively. The frequency of overweight and obesity in the first study was considerably higher than the estimated prevalence in Goiânia, while the second was similar for obesity but with higher occurrence of overweight cats. In this context, the discrepancies among the four studies conducted on feline obesity prevalence may be attributed to methodological differences, as two were based on hospital-attending populations and two employed a census-based home visit model. Although the latter is theoretically more representative and reliable, it is inherently influenced by the regional

**Table 2. Binary logistic regression analysis of variables statistically significant in the Kruskal-Wallis test associated with overweight and obese body condition of owned cats in Goiânia, Brazil.**

| Characteristic | Ideal (BCS 5) | | Overweight and obese (BCS ≥ 6) | | Total | | OR | CI-95% | p-value* |
|---|---|---|---|---|---|---|---|---|---|
| | N | % | N | % | N | % | | | |
| **Age range** | | | | | | | | | |
| 1 to 6 years | 64 | (64.6) | 35 | (35.4) | 99 | (100) | 3.5 | 1.4-9.1 | 0.008 |
| 7 to 10 years | 8 | (34.8) | 15 | (65.2) | 23 | (100) | | | |
| **Reproductive status** | | | | | | | | | |
| Neutered | 63 | (56.3) | 49 | (43.7) | 112 | (100) | 0.459 | 0.15-1.4 | 0.161 |
| Intact | 14 | (73.7) | 5 | (26.3) | 19 | (100) | | | |
| **Physical activity** | | | | | | | | | |
| Active | 55 | (67.9) | 26 | (32.1) | 81 | (100) | 2.7 | 1.3-5.6 | 0.008 |
| Inactive | 22 | (44.0) | 28 | (56.0) | 50 | (100) | | | |
| **Vaccinated** | | | | | | | | | |
| Yes | 66 | (57.9) | 48 | (42.1) | 114 | (100) | 0.75 | 0.3-2.2 | 0.595 |
| No | 11 | (64.7) | 6 | (35.3) | 17 | (100) | | | |
| **Vaccination frequency** | | | | | | | | | |
| Annual | 47 | (59.5) | 32 | (40.5) | 79 | (100) | 1.2 | 0.5-2.8 | 0.604 |
| Overdue | 19 | (54.3) | 16 | (45.7) | 35 | (100) | | | |
| **Has been assessed by a veterinarian** | | | | | | | | | |
| Yes | 21 | (70.0) | 9 | (30.0) | 30 | (100) | 0.5 | 0.2-1.3 | 0.159 |
| No | 56 | (55.4) | 45 | (44.6) | 101 | (100) | | | |
| **Frequency of visits to a veterinarian** | | | | | | | | | |
| Annual | 23 | (54.8) | 19 | (45.2) | 42 | (100) | 1 | 0.5-2.2 | 0.981 |
| Only when sick | 33 | (55.0) | 27 | (45.0) | 60 | (100) | | | |
| **Disease status reported by the owner** | | | | | | | | | |
| Yes | 23 | (69.7) | 10 | (30.3) | 33 | (100) | 2.8 | 0.9-8.5 | 0.076 |
| No | 10 | (45.5) | 12 | (54.5) | 22 | (100) | | | |
| **Type of household** | | | | | | | | | |
| House | 76 | (62.3) | 46 | (37.7) | 122 | (100) | 13.2 | 1.6-109.1 | 0.017 |
| Apartament | 1 | (11.1) | 8 | (88.9) | 9 | (100) | | | |
| **Presence of other cat(s) in the household** | | | | | | | | | |
| Yes | 65 | (61.9) | 40 | (38.1) | 105 | (100) | 1.9 | 0.8-4.5 | 0.148 |
| No | 12 | (46.2) | 14 | (53.8) | 26 | (100) | | | |
| **Presence of dog(s) in the household** | | | | | | | | | |
| Yes | 29 | (61.7) | 18 | (38.3) | 47 | (100) | 1.2 | 0.6-2.5 | 0.611 |
| No | 48 | (57.1) | 36 | (42.9) | 84 | (100) | | | |
| **Number of cats in the household** | | | | | | | | | |
| 1–4 | 31 | (43.7) | 40 | (56.3) | 71 | (100) | 0.2 | 0.1-0.5 | <0.0001 |
| 5 or more | 46 | (76.7) | 14 | (23.3) | 60 | (100) | | | |
| **Education Level** | | | | | | | | | |
| No formal education or up to middle school | 21 | (84.0) | 4 | (16.0) | 25 | (100) | 4.7 | 1.5-14.6 | 0.008 |
| High school to postgraduate education | 56 | (52.8) | 50 | (47.2) | 106 | (100) | | | |
| **Owner's occupation** | | | | | | | | | |
| Home | 50 | (68.5) | 23 | (31.5) | 73 | (100) | 2.5 | 1.2-5.1 | 0.012 |
| Outside of home | 27 | (46.5) | 31 | (53.5) | 58 | (100) | | | |
| **Household income** | | | | | | | | | |

*(Continued)*

**Table 2.** (Continued)

| Characteristic | Ideal (BCS 5) | | Overweight and obese (BCS ≥ 6) | | Total | | OR | CI-95% | p-value* |
|---|---|---|---|---|---|---|---|---|---|
| | N | % | N | % | N | % | | | |
| Up to $ 670 | 52 | (65.0) | 28 | (35.0) | 80 | (100) | 1.9 | 0.9-3,9 | 0.072 |
| More than $ 670 | 25 | (49.0) | 26 | (51.0) | 51 | (100) | | | |
| **City region** | | | | | | | | | |
| Other regions | 56 | (69.1) | 25 | (30.9) | 81 | (100) | 0.3 | 0.1-0.7 | 0.003 |
| South/East/North | 21 | (42.0) | 29 | (58.0) | 50 | (100) | | | |
| **Perception of the owner's eating habits** | | | | | | | | | |
| Normal/In excess | 71 | (62.8) | 42 | (37.2) | 113 | (100) | 0.3 | 0.1-0.8 | 0.023 |
| Low amounts | 6 | (33.3) | 12 | (66.7) | 18 | (100) | | | |

BCS: Body Condition Score; OR: odds ratio; CI-95%: 95% confidence interval. *P-value obtained by the binary logistic regression.

characteristics of the sampled population. In the present study, for instance, the number of veterinary visits showed a statistically significant association with obesity prevalence, as identified by the Kruskal-Wallis test.

Regarding the sanitary aspects evaluated, differences in BCS distribution were observed between vaccination status, vaccination frequency, veterinary visits, and their frequency, as well as the diseases status reported by the owner (Table 3). Lack of or delayed vaccination, as well as not visiting the veterinarian, were associated with higher prevalence of underweight cats, which may have been due to a higher frequency of diseases in these groups. In our sample, this theory was supported by the higher prevalence of underweight cats that, according to the owners, were ill, although none of these sanitary characteristics were considered risk factors for obesity (Table 2). Similarly, it can be hypothesized that animals with more frequent veterinary visits tend to receive greater overall care from their owners, which may include the provision of higher-quality diets—typically characterized by greater digestibility and caloric density. These are likely indoor cats with limited space for physical activity, that are neutered and have greater longevity. Therefore, these factors may explain why the frequency of veterinary visits is associated with the distribution of body condition scores and the observed differences in the prevalence of overweight and obesity across this data and other Brazilian studies [16–18].

Chiang et al. [17] reported a higher prevalence of BCS between 1 and 5 in cats with immune/infectious diseases. In the same study, dental, orthopedic, and urinary diseases were positively associated with overweight and obesity. Lund et al. [10] found a higher prevalence of overweight or obese cats with oral cavity and urinary tract diseases. In our study, however, the specific diseases reported by owners did not influence the BCS distribution of the animals.

None of the feeding management characteristics were statistically associated with the cats' BCS (Table 4). Supporting these findings, another study conducted in Brazil with owned cats in a smaller city also found no correlation between feeding frequency or type of food provided and obesity development [16]. Studies conducted in New Zealand [6] and France [10] also did not identify feeding management factors as risk factors for obesity. On the other hand, food quantity [28], free access to food [7,17], commercial dry diets [27,28], and the lack of food portion control [27] have been described as risk factors for the development of overweight or obesity in cats.

The type of household had a statistically significant influence on cats' BCS (p = 0.003). The prevalence of overweight or obese cats was higher in the apartment group (Table 5). Living in an apartment was identified as a risk factor for overweight and obesity (OR = 13.2) (Table 2). Partial or total restriction to outdoor environments has also been reported as a risk factor for obesity in cats in other studies [14,27,28].

**Table 3. Distribution of body condition score (BCS) frequencies according to the sanitary characteristics of owned cats in Goiânia, Brazil.**

| Sanitary Characteristics | Underweight (BCS 1–4) | | Ideal (BCS 5) | | Overweight (BCS 6–7) | | Obese (BCS 8–9) | | Total | | p-value* |
|---|---|---|---|---|---|---|---|---|---|---|---|
| | N | % | N | % | N | % | N | % | N | % | |
| **Vaccinated** | | | | | | | | | | | |
| Yes | 36 | (24.0) | 66 | (44.0) | 33 | (22.0) | 15 | (10.0) | 150 | (79.8) | 0.002 |
| No | 21 | (55.3) | 11 | (28.9) | 5 | (13.2) | 1 | (2.6) | 38 | (20.2) | |
| **Vaccination frequency** | | | | | | | | | | | |
| Annual | 20 | (20.2) | 47 | (47.4) | 23 | (23.3) | 9 | (9.1) | 99 | (52.7) | 0.003 |
| Overdue | 16 | (31.4) | 19 | (37.2) | 10 | (19.6) | 6 | (11.8) | 51 | (27.1) | |
| Never | 21 | (55.3) | 11 | (28.9) | 5 | (13.2) | 1 | (2.6) | 38 | (20.2) | |
| **Vaccination in a public rabies campaign** | | | | | | | | | | | |
| No | 36 | (30.2) | 48 | (40.3) | 24 | (20.2) | 11 | (9.2) | 119 | (63.3) | 0.972 |
| Yes | 21 | (30.4) | 29 | (42.0) | 14 | (20.3) | 5 | (7.3) | 69 | (36.7) | |
| **Vaccinated by veterinarian** | | | | | | | | | | | |
| No | 24 | (38.7) | 22 | (35.5 | 10 | (16.1) | 6 | (9.7) | 62 | (33.0) | 0.294 |
| Yes | 33 | (26.2) | 55 | (43.7) | 28 | (22.2) | 10 | (7.9) | 126 | (67.0) | |
| **Has been assessed by a veterinarian** | | | | | | | | | | | |
| No | 26 | (46.4) | 21 | (37.5) | 6 | (10.7) | 3 | (5.4) | 56 | (29.8) | 0.009 |
| Yes | 31 | (23.5) | 56 | (42.4) | 32 | (24.2) | 13 | (9.9) | 132 | (70.2) | |
| **Frequency of visits to a veterinarian** | | | | | | | | | | | |
| Annual | 14 | (25.0) | 23 | (41.1) | 14 | (25.0) | 5 | (8.9) | 56 | (29.8) | 0.035 |
| Only when sick | 17 | (22.1) | 33 | (42.9) | 18 | (23.3) | 9 | (11.7) | 77 | (41.0) | |
| Never | 26 | (47.3) | 21 | (38.2) | 6 | (10.9) | 2 | (3.6) | 55 | (29.2) | |
| **Disease reported by the owner** | | | | | | | | | | | |
| No disease | 32 | (24.6) | 54 | (41.6) | 32 | (24.6) | 12 | (9.2) | 130 | (69.1) | 0.129 |
| Infectious or parasitic disease | 20 | (54.0) | 14 | (37.9) | 2 | (5.4) | 1 | (2.7) | 37 | (19.7) | |
| Orthopedic disease | 3 | (27.3) | 5 | (45.5) | 2 | (18.1) | 1 | (9.1) | 11 | (5.9) | |
| Urinary disease | 1 | (25.0) | 1 | (25.0) | 2 | (50.0) | 0 | (0.0) | 4 | (2.1) | |
| Neoplasia | 0 | (0.0) | 2 | (100) | 0 | (0.0) | 0 | (0.0) | 2 | (1.1) | |
| Intoxication | 1 | (50.0) | 0 | (0.0) | 0 | (0.0) | 1 | (50.0) | 2 | (1.1) | |
| Gastritis | 0 | (0.0) | 1 | (100) | 0 | (0.0) | 0 | (0.0) | 1 | (0.5) | |
| Diabetes mellitus | 0 | (0.0) | 0 | (0.0) | 0 | (0.0) | 1 | (100) | 1 | (0.5) | |
| **Disease status reported by the owner** | | | | | | | | | | | |
| Yes | 25 | (43.1) | 23 | (39.7) | 6 | (10.3) | 4 | (6.9) | 58 | (30.8) | 0.043 |
| No | 8 | (26.7) | 10 | (33.3) | 10 | (33.3) | 2 | (6.7) | 30 | (69.2) | |
| **FIV and/or FeLV reported by owners** | | | | | | | | | | | |
| Yes | 8 | (47.0) | 7 | (41.2) | 1 | (5.9) | 1 | (5.9) | 17 | (19.3) | 0.215 |
| No | 25 | (35.2) | 26 | (36.6) | 15 | (21.1) | 5 | (7.1) | 71 | (80.7) | |

BCS: Body Condition Score; *p value obtained by the Kruskal-Wallis test.

The presence of other pets in the household influenced the distribution of the cats' BCS (Table 5). However, when analyzed through binary logistic regression, it was not considered a risk factor for overweight or obesity, despite the higher prevalence of overweight or obese cats in homes without other cats (Table 2).

When BCS was evaluated based on the number of cats in the household, it was observed that the presence of at least five cats in the household acted as a protective factor against overweight or obesity (p < 0.001; Table 2). In the study

**Table 4. Distribution of body condition score (BCS) according to the feeding management of owned cats in Goiânia, Brazil.**

| Feeding management | Underweight (BCS 1–4) | | Ideal (BCS 5) | | Overweight (BCS 6–7) | | Obese (BCS 8–9) | | Total | | p-value* |
|---|---|---|---|---|---|---|---|---|---|---|---|
| | N | % | N | % | N | % | N | % | N | % | |
| **Type of diet** | | | | | | | | | | | |
| Commercial diets (wet and dry) | 33 | (25.2) | 55 | (42.0) | 29 | (22.1) | 14 | (10.7) | 131 | (69.7) | 0.064 |
| Commercial diets (wet and dry) and homemade diet | 23 | (41.1) | 22 | (39.3) | 9 | (16.1) | 2 | (3.5) | 56 | (29.8) | |
| Only homemade diet | 1 | (100) | 0 | (0.0) | 0 | (0.0) | 0 | (0.0) | 1 | (0.5) | |
| **Feeding frequency** | | | | | | | | | | | |
| Once a day | 0 | (0.0) | 0 | (0.0) | 1 | (100) | 0 | (0.0) | 1 | (0.5) | 0.640 |
| Twice a day | 17 | (37.8) | 20 | (44.4) | 4 | (8.9) | 4 | (8.9) | 45 | (24.0) | |
| Three times a day | 6 | (17.1) | 17 | (48.6) | 9 | (25.7) | 3 | (8.6) | 35 | (18.6) | |
| *Ad libitum* | 34 | (31.8) | 40 | (37.4) | 24 | (22.4) | 9 | (8.4) | 107 | (56.9) | |
| **Method of quantification of daily food intake** | | | | | | | | | | | |
| Measuring cup | 4 | (23.5) | 7 | (41.2) | 5 | (29.4) | 1 | (5.9) | 17 | (9.0) | 0.750 |
| Does not quantify | 53 | (31.0) | 70 | (40.9) | 33 | (19.3) | 15 | (8.8) | 171 | (91.0) | |
| **Criteria to determine daily food intake** | | | | | | | | | | | |
| Veterinarian | 0 | (0.0) | 0 | (0.0) | 1 | (100) | 0 | (0.0) | 1 | (0.5) | 0.731 |
| Owner choice | 4 | (25.0) | 7 | (43.7) | 4 | (25.0) | 1 | (6.3) | 16 | (8.5) | |
| No information | 53 | (31.0) | 70 | (40.9) | 33 | (19.3) | 15 | (8.8) | 171 | (91.0) | |
| **Gives treats** | | | | | | | | | | | |
| Yes | 37 | (31.9) | 47 | (40.5) | 24 | (20.7) | 8 | (6.9) | 116 | (61.7) | 0.749 |
| No | 20 | (27.8) | 30 | (41.7) | 14 | (19.4) | 8 | (11.1) | 72 | (38.3) | |
| **Type of treats** | | | | | | | | | | | |
| Cat treats | 13 | (28.3) | 18 | (39.1) | 11 | (23.9) | 4 | (8.7) | 46 | (24.4) | 0.725 |
| Human food | 6 | (40.0) | 6 | (40.0) | 2 | (13.3) | 1 | (6.7) | 15 | (8.0) | |
| Cat treats and human food | 18 | (32.7) | 23 | (41.8) | 11 | (20.0) | 3 | (5.5) | 55 | (29.3) | |
| Does not give treats | 20 | (27.8) | 30 | (41.7) | 14 | (19.4) | 8 | (11.1) | 72 | (38.3) | |

BCS: Body Condition Score; *p value obtained by the Kruskal-Wallis test.

conducted by Cave et al. [12], the number of cats or the presence of dogs in the household did not influence the prevalence of overweight or obesity. Similarly, Colliard et al. [10] did not find a correlation between overweight or obesity and living with other animals. The number of people in the household or the presence of children or elderly individuals did not influence the distribution of BCS in the study cats.

Unlike sex, age group, and work hours, the owners' education level influenced the cats' BCS (p = 0.002). A higher prevalence of obese cats was observed among owners with higher education levels (Table 6). Owners with education from high school to postgraduate levels were considered a risk factor for the development of overweight or obesity in the study cats (Table 2).

Household income influenced the cats' BCS (Table 6). Although household income was not identified as a risk factor for overweight and obesity when analyzed in binary form (Table 2), an increasing prevalence of obese cats was observed as household income increased (Table 6).

The Southern region of Goiânia has a population with a high average household income, according to the literature [29], and in our study, most of the owners living in apartments were concentrated in this area. These factors may explain the high prevalence of obese cats in the southern region. In contrast, the northwestern region of Goiânia, which has a higher concentration of lower-income residents [29], showed the highest prevalence of underweight cats. Excluding the

**Table 5. Distribution of body condition score (BCS) frequencies according to household characteristics of owned cats in Goiânia, Brazil.**

| Household characteristics | Underweight (BCS 1–4) | | Ideal (BCS 5) | | Overweight (BCS 6–7) | | Obese (BCS 8–9) | | Total | | p-value* |
|---|---|---|---|---|---|---|---|---|---|---|---|
| | N | % | N | % | N | % | N | % | N | % | |
| **Type of household** | | | | | | | | | | | |
| House | 56 | (31.5) | 76 | (42.7) | 33 | (18.5) | 13 | (7.3) | 178 | (94.7) | 0.003 |
| Apartament | 1 | (10.0) | 1 | (10.0) | 5 | (50.0) | 3 | (30.0) | 10 | (5.3) | |
| **Other pets in the household** | | | | | | | | | | | |
| Cats | 29 | (29.6) | 41 | (41.8) | 20 | (20.4) | 8 | (8.2) | 98 | (52.1) | 0.016 |
| Dogs | 0 | (0.0) | 5 | (45.4) | 4 | (36.4) | 2 | (18.2) | 11 | (5.8) | |
| Dogs and cats | 25 | (41.0) | 24 | (39.3) | 8 | (13.1) | 4 | (6.6) | 61 | (32.5) | |
| No other animals | 3 | (16.7) | 7 | (38.9) | 6 | (33.3) | 2 | (11.1) | 18 | (9.6) | |
| **Number of cats in the household** | | | | | | | | | | | |
| 1 | 3 | (10.3) | 12 | (41.4) | 10 | (34.5) | 4 | (13.8) | 29 | (15.4) | <0.001 |
| 2 | 5 | (20.8) | 6 | (25.0) | 7 | (29.2) | 6 | (25.0) | 24 | (12.8) | |
| 3 or 4 | 10 | (27.8) | 13 | (36.1) | 9 | (25.0) | 4 | (11.1) | 36 | (19.1) | |
| 5–10 | 18 | (34.6) | 25 | (48.1) | 7 | (13.4) | 2 | (3.9) | 52 | (27.7) | |
| 15 or more | 21 | (44,7) | 21 | (44.7) | 5 | (10,6) | 0 | (0,0) | 47 | (25,0) | |
| **Number of people in the household** | | | | | | | | | | | |
| 1 | 12 | (33.3) | 14 | (38.9) | 6 | (16.7) | 4 | (11.1) | 36 | (19.2) | 0.107 |
| 2 | 15 | (36.6) | 14 | (34.1) | 5 | (12.2) | 7 | (17.1) | 41 | (21.8) | |
| 3 | 8 | (22.9) | 13 | (37.1) | 12 | (34.3) | 2 | (5.7) | 35 | (18.6) | |
| 4 | 11 | (30.6) | 18 | (50.0) | 5 | (13.9) | 2 | (5.5) | 36 | (19.1) | |
| 5 | 5 | (27.8) | 9 | (50.0) | 3 | (16.7) | 1 | (5.5) | 18 | (9.6) | |
| 6 | 6 | (31.6) | 8 | (42.1) | 5 | (26.3) | 0 | (0.0) | 19 | (10.1) | |
| 10 | 0 | (0.0) | 1 | (33.3) | 2 | (66.7) | 0 | (0.0) | 3 | (1.6) | |
| **Children in the household** | | | | | | | | | | | |
| Yes | 16 | (33.3) | 20 | (41.7) | 10 | (20.8) | 2 | (4.2) | 48 | (25.5) | 0.652 |
| No | 41 | (29.3) | 57 | (40.7) | 28 | (20.0) | 14 | (10.0) | 140 | (74.5) | |
| **Elderly people in the household** | | | | | | | | | | | |
| Yes | 40 | (33.3) | 51 | (42.5) | 21 | (17.5) | 8 | (6.7) | 120 | (63.8) | 0.296 |
| No | 17 | (25.0) | 26 | (38.2) | 17 | (24.0) | 8 | (11.8) | 68 | (36.2) | |

BCS: Body Condition Score; *p value obtained by the Kruskal-Wallis test.

southern, eastern, and northern regions of the city, living in other areas was considered a protective factor against overweight and obesity in the owned cats in Goiânia (Table 2).

The owner's occupation influenced the cats' BCS (p = 0.028). Working outside the home was identified as a risk factor for overweight and obesity (OR = 2.5). Although age group did not influence BCS in this study, Colliard et al. [10] found an increased risk of overweight in cats owned by individuals aged 41–60 years.

Owners' habits related to physical activity, as well as the consumption of fried foods, fruits, vegetables, or snacks did not influence the cats' BCS (Table 7). In dogs, however, the unhealthy habit of consuming snacks by owners has been identified as a risk factor for the development of overweight and obesity in their pets [1].

The owners' perceptions of their own eating habits were different among different BCS (Table 7). Cats from owners who perceived themselves as eating normal or excessive amounts had a lower prevalence of overweight or obesity (Table 2), with this perception serving as a protective factor. These owners, who considered their food intake to be normal or excessive,

**Table 6. Distribution of body condition score (BCS) frequencies according to the intrinsic and socioeconomic characteristics of cat owners in Goiânia, Brazil.**

| Characteristics | Underweight (BCS 1–4) | | Ideal (BCS 5) | | Overweight (BCS 6–7) | | Obese (BCS 8–9) | | Total | | p-value* |
|---|---|---|---|---|---|---|---|---|---|---|---|
| | N | % | N | % | N | % | N | % | N | % | |
| **Gender** | | | | | | | | | | | |
| Male | 8 | (25.8) | 15 | (48.4) | 6 | (19.3) | 2 | (6.5) | 31 | (16.5) | 0.815 |
| Female | 49 | (31.1) | 62 | (39.5) | 32 | (20.4) | 14 | (9.0) | 157 | (83.5) | |
| **Age range** | | | | | | | | | | | |
| 18 to 34 years | 9 | (27.3) | 14 | (42.4) | 8 | (24.2) | 2 | (6.1) | 33 | (17.6) | 0.180 |
| 35 to 59 years | 18 | (23.7) | 30 | (39.5) | 20 | (26.3) | 8 | (10.5) | 76 | (40.4) | |
| ≥ 60 | 30 | (37.9) | 33 | (41.8) | 10 | (12.7) | 6 | (7.6) | 79 | (42.0) | |
| **Education level** | | | | | | | | | | | |
| Did not attend school | 0 | (0.0) | 4 | (80.0) | 1 | (20.0) | 0 | (0.0) | 5 | (2.7) | 0.002 |
| Elementary School | 14 | (60.9) | 7 | (30.4) | 2 | (8.7) | 0 | (0.0) | 23 | (12.2) | |
| Middle School | 3 | (21.4) | 10 | (71.4) | 1 | (7.2) | 0 | (0.0) | 14 | (7.5) | |
| High School | 23 | (31.9) | 28 | (38.9) | 17 | (23.6) | 4 | (5.6) | 72 | (38.3) | |
| Collage/University | 13 | (22.8) | 23 | (40.3) | 12 | (21.1) | 9 | (15.8) | 57 | (30.3) | |
| Postgraduate | 4 | (23.5) | 5 | (29.4) | 5 | (29.4) | 3 | (17.7) | 17 | (9.0) | |
| **Owner's occupation** | | | | | | | | | | | |
| Home | 37 | (33.6) | 50 | (45.4) | 18 | (16.4) | 5 | (4.6) | 110 | (58.5) | 0.028 |
| Outside of home | 20 | (25.6) | 27 | (34.7) | 20 | (25.6) | 11 | (14.1) | 78 | (41.5) | |
| **Work hours** | | | | | | | | | | | |
| Up to10h/week | 35 | (32.7) | 49 | (45.8) | 18 | (16.8) | 5 | (4.7) | 107 | (56.9) | 0.216 |
| 11h to 30h/week | 5 | (20.8) | 7 | (29.2) | 8 | (33.3) | 4 | (16.7) | 24 | (12.8) | |
| 30h to 40h/week | 10 | (40.0) | 5 | (20.0) | 6 | (24.0) | 4 | (16.0) | 25 | (13.3) | |
| >40 h/week | 7 | (21.9) | 16 | (50.0) | 6 | (18.7) | 3 | (9.4) | 32 | (17.0) | |
| **Household income** | | | | | | | | | | | |
| Up to $ 225 | 16 | (59.3) | 9 | (33.3) | 2 | (7.4) | 0 | (0.0) | 27 | (14.4) | 0.009 |
| From $ 225 to $ 670 | 22 | (24.2) | 43 | (47.2) | 20 | (22.0) | 6 | (6.6) | 91 | (48.4) | |
| From $ 670 to $ 1.790 | 16 | (36.4) | 13 | (29.5) | 10 | (22.7) | 5 | (11.4) | 44 | (23.4) | |
| More than $ 1.790 | 3 | (11.5) | 12 | (46.2) | 6 | (23.1) | 5 | (19.2) | 26 | (13.8) | |
| **City region** | | | | | | | | | | | |
| Central | 23 | (44.3) | 18 | (34.6) | 10 | (19.2) | 1 | (1.9) | 52 | (27.7) | 0.001 |
| East | 2 | (10.5) | 7 | (36.8) | 9 | (47.4) | 1 | (5.3) | 19 | (10.1) | |
| Northwest | 16 | (51.6) | 12 | (38.7) | 3 | (9.7) | 0 | (0.0) | 31 | (16.5) | |
| North | 1 | (10.0) | 4 | (40.0) | 2 | (20.0) | 3 | (30.0) | 10 | (5.3) | |
| West | 2 | (10.5) | 12 | (63.2) | 4 | (21.0) | 1 | (5.3) | 19 | (10.1) | |
| South-west | 12 | (37.5) | 14 | (43.7) | 4 | (12.5) | 2 | (6.3) | 32 | (17.0) | |
| South | 1 | (4.0) | 10 | (40.0) | 6 | (24.0) | 8 | (32.0) | 25 | (13.3) | |

BCS: Body Condition Score; *p value obtained by the Kruskal-Wallis test.

appear to have a more accurate perception of their own diet, which may reflect into providing a more appropriate amounts of food for their cats. In contrast, of the 14 owners who claimed to eat low amounts, 8 (57.1%) were overweight or obese, and their cats' food was not quantified, suggesting a less accurate perception of their own eating habits, which may also be reflected in their pets' diet.

**Table 7. Distribution of body condition score (BCS) frequencies according to the lifestyle habits of cat owners in Goiânia, Brazil.**

| Characteristics | Underweight (BCS 1–4) | | Ideal (BCS 5) | | Overweight (BCS 6–7) | | Obese (BCS 8–9) | | Total | | p-value* |
|---|---|---|---|---|---|---|---|---|---|---|---|
| | N | % | N | % | N | % | N | % | N | % | |
| **Physical activity** | | | | | | | | | | | |
| Yes | 21 | (25.0) | 39 | (46.4) | 15 | (17.9) | 9 | (10.7) | 84 | (44.7) | 0.285 |
| No | 36 | (34.6) | 38 | (36.6) | 23 | (22.1) | 7 | (6.7) | 104 | (55.3) | |
| **Frequency of exercise** | | | | | | | | | | | |
| Does not exercise | 36 | (34.3) | 39 | (37.1) | 23 | (21.9) | 7 | (6.7) | 105 | (55.8) | 0.262 |
| Once a week | 3 | (16.7) | 11 | (61.1) | 2 | (11.1) | 2 | (11.1) | 18 | (9.6) | |
| Three times a week | 6 | (24.0) | 7 | (28.0) | 5 | (20.0) | 7 | (28.0) | 25 | (13.3) | |
| More than three times a week | 5 | (41.7) | 6 | (50.0) | 1 | (8.3) | 0 | (0.0) | 12 | (6.4) | |
| Daily | 7 | (25.0) | 14 | (50.0) | 7 | (25.0) | 0 | (0.0) | 28 | (14.9) | |
| **Perception of own eating habits** | | | | | | | | | | | |
| In excess | 6 | (24.0) | 15 | (60.0) | 2 | (8.0) | 2 | (8.0) | 25 | (13.3) | 0.007 |
| Normal | 42 | (30.9) | 56 | (41.2) | 25 | (18.4) | 13 | (9.5) | 136 | (72.3) | |
| Low amounts | 9 | (33.3) | 6 | (22.2) | 11 | (40.8) | 1 | (3.7) | 27 | (14.4) | |
| **Fried food consumption** | | | | | | | | | | | |
| Healthy | 37 | (26.4) | 59 | (42.2) | 30 | (21.4) | 14 | (10.0) | 140 | (74.5) | 0.193 |
| Unhealthy | 20 | (41.7) | 18 | (37.5) | 8 | (16.7) | 2 | (4.1) | 48 | (25.5) | |
| **Fruit consumption** | | | | | | | | | | | |
| Healthy | 52 | (32.1) | 66 | (40.7) | 32 | (19.8) | 12 | (7.4) | 162 | (86.2) | 0.389 |
| Unhealthy | 5 | (19.2) | 11 | (42.3) | 6 | (23.1) | 4 | (15.4) | 26 | (13.8) | |
| **Vegetable consumption** | | | | | | | | | | | |
| Healthy | 49 | (28.3) | 72 | (41.6) | 36 | (20.8) | 16 | (9.3) | 173 | (92.0) | 0.185 |
| Unhealthy | 8 | (53.3) | 5 | (33.3) | 2 | (13.3) | 0 | (0.0) | 15 | (8.0) | |
| **Snack consumption** | | | | | | | | | | | |
| Healthy | 42 | (30.4) | 57 | (41.3) | 31 | (22.5) | 8 | (5.8) | 138 | (73.4) | 0.123 |
| Unhealthy | 15 | (30.0) | 20 | (40.0) | 7 | (14.0) | 8 | (16.0) | 50 | (26.6) | |

BCS – Body Condition Score; *p value obtained by the Kruskal-Wallis test.

The anthropometric measurements taken from the owners resulted in the body mass index (BMI) and calculations for waist/hip, waist/height, and abdominal circumference ratios. None of these characteristics were related to the cats' BCS (Table 8). In Brazil, BMI and the same risk factors were evaluated for dog owners, and similarly, no statistical relationships were found with the dogs' BCS [1]. On the other hand, according to Loftus and Wakshlag [3], the relationship between the obesity of owners and their pets is better described for dogs than for cats. A study on the association between the BMI of owners and the overweight of their dogs and cats was conducted in the Netherlands, with a significant positive correlation found only for dogs [30].

The underestimation of the animals' body condition score (BCS) by their owners has been described as a risk factor for the development of overweight and obesity in cats [6,10,12,17]. According to Cave et al. [12], from a feline population health perspective, the association between obesity and the owner's perception of their cat's body condition suggests that more attention should be directed toward educating owners about the ideal body condition of a cat, rather than focusing educational efforts on changing feeding patterns or types of food for cats. Consistent with the literature [31], our results showed only a slight agreement (kappa = 0.177; p < 0.001; Table 9) between the cats' BCS and the owners' perceptions of these BCS. Interestingly, 71.1% and 75.0% of owners underestimated the BCS of overweight or obese cats, respectively (Table 9), classifying overweight cats as ideal (68.5%) and obese cats as overweight (50%) (Table 9).

 

**Table 8. Distribution of body condition score (BCS) frequencies according to the anthropometric measurements of cat owners in Goiânia, Brazil.**

| Characteristics | Underweight (BCS 1–4) | | Ideal (BCS 5) | | Overweight (BCS 6–7) | | Obese (BCS 8–9) | | Total | | p-value* |
|---|---|---|---|---|---|---|---|---|---|---|---|
| | N | % | N | % | N | % | N | % | N | % | |
| **Owner's body mass index** | | | | | | | | | | | |
| Underweight (<18.5) | 2 | (12.5) | 9 | (56.2) | 3 | (18.8) | 2 | (12.5) | 16 | (8.5) | 0.800 |
| Eutrophic (18.6–24.9) | 27 | (31.8) | 35 | (41.2) | 17 | (20.0) | 6 | (7.0) | 85 | (45.2) | |
| Overweight (25.0–29.9) | 20 | (36.4) | 20 | (36.4) | 12 | (21.8) | 3 | (5.4) | 55 | (29.3) | |
| Obese (≥30.0) | 8 | (25.0) | 13 | (40.6) | 6 | (18.8) | 5 | (15.6) | 32 | (17.0) | |
| **Owner waist/hip ratio** | | | | | | | | | | | |
| High risk | 25 | (37.9) | 25 | (37.9) | 11 | (16.6) | 5 | (7.6) | 66 | (35.1) | 0.083 |
| Moderate risk | 27 | (32.5) | 34 | (41.0) | 17 | (20.5) | 5 | (6.0) | 83 | (44.2) | |
| Low risk | 5 | (12.8) | 18 | (46.1) | 10 | (25.7) | 6 | (15.4) | 39 | (20.7) | |
| **Owner waist/height** | | | | | | | | | | | |
| At risk | 31 | (36.0) | 34 | (39.5) | 14 | (16.3) | 7 | (8.2) | 86 | (45.7) | 0.389 |
| No at risk | 26 | (25.5) | 43 | (42.2) | 24 | (23.5) | 9 | (8.8) | 102 | (54.3) | |
| **Owner abdominal circumference** | | | | | | | | | | | |
| At risk | 48 | (31.8) | 63 | (41.7) | 29 | (19.2) | 11 | (7.3) | 151 | (80.3) | 0.497 |
| No at risk | 9 | (24.3) | 14 | (37.9) | 9 | (24.3) | 5 | (13.5) | 37 | (19.7) | |

BCS – Body condition score; *p value obtained by the Kruskal-Wallis test.

**Table 9. Comparison of agreement between body condition scores (BCS) determined by owner and veterinarians in a population of owned cats in Goiânia, Brazil.**

| Owner perception of the cats' BCS | Underweight (BCS 1–4) | | Ideal (BCS 5) | | Overweight (BCS 6–7) | | Obese (BCS 8–9) | | p-value* | K# |
|---|---|---|---|---|---|---|---|---|---|---|
| | N | % | N | % | N | % | N | % | | |
| Underweight | 9 | (15,8) | 1 | (1,3) | 1 | (2,6) | 0 | (0,0) | <0,001 | 0,177 |
| Ideal | 45 | (78,9) | 66 | (85,7) | 26 | (68,5) | 4 | (25,0) | | |
| Overweight | 3 | (5,3) | 10 | (13,0) | 10 | (26,3) | 8 | (50,0) | | |
| Obese | 0 | (0,0) | 0 | (0,0) | 1 | (2,6) | 4 | (25,0) | | |
| Total | 57 | (100) | 77 | (100) | 38 | (100) | 16 | (100) | | |

BCS – Body Condition Score; *p value obtained by the kappa test; #K correspondent to kappa test (inter-rater agreement).

Despite being conducted with strict methodological criteria, this study had some limitations. Visits to the households were made during business hours, which likely increased the number of households with absent residents that were not sampled, as well as the sampling of owners who worked from home, were retired, or unemployed. Another limitation relates to the behavior of domestic cats. Many animals were not found by their owners at the time of the visit or were aggressive and did not allow the necessary assessments to be conducted. Furthermore, due to defensive behavior or fear of being judged, some owners may have answered questions about their own lifestyle habits or the management of their pets by stating what they believed to be correct, rather than reflecting the reality, causing unintended bias. Despite the small limitations inherent in applying questionnaires in field research, our results consistently provide, for the first time, an epidemiological diagnosis of obesity in domestic cats in a metropolitan area in Latin America.

## Conclusions

The prevalence of feline overweight and obese in Goiânia, Brazil, was estimated at 28.7%, with 20.2% of the cats being overweight and 8.5% obese. Factors associated with the development of overweight and obesity included the cat's age (between 7 and 10 years), low physical activity, living in apartment, the owner's education level (from high school to postgraduate), and the owner working outside the home. The presence of five or more cats in the same household, the location of the household in specific areas of the city, and the owner reporting that their own food intake is normal or excessive were protective factors against the development of overweight and obesity in the studied cat population. The BCS of overweight or obese cats was underestimated by 71.1% and 75%, respectively. This was the first study to provide important insights into the prevalence and risk factors of feline obesity in a metropolitan area of Latin America.

## Acknowledgments

The authors would like to thank all the owners who volunteered to participate in this study. We are also grateful to the National Council for Scientific and Technological Development (CNPq) for the scholarships granted.

## Author contributions

**Conceptualization:** Danilo Conrado Silva, Fabio Alves Teixeira, Mariana Yukari Hayasaki Porsani.

**Data curation:** Danilo Conrado Silva, Aparecido Divino da Cruz, Lysa Bernardes Minasi, Vitória Alvarenga Nunes.

**Formal analysis:** Aparecido Divino da Cruz, Vitória Alvarenga Nunes, Alex Silva da Cruz.

**Investigation:** Danilo Conrado Silva, Fabio Alves Teixeira, Aparecido Divino da Cruz, Erika Figueiredo Pereira, Lysa Bernardes Minasi, Alex Silva da Cruz.

**Methodology:** Danilo Conrado Silva, Fabio Alves Teixeira, Aparecido Divino da Cruz, Erika Figueiredo Pereira, Klayto José Gonçalves dos Santos, Lysa Bernardes Minasi, Vitória Alvarenga Nunes, Mariana Yukari Hayasaki Porsani.

**Project administration:** Danilo Conrado Silva.

**Software:** Aparecido Divino da Cruz, Alex Silva da Cruz.

**Supervision:** Fabio Alves Teixeira, Mariana Yukari Hayasaki Porsani.

**Validation:** Danilo Conrado Silva, Fabio Alves Teixeira, Aparecido Divino da Cruz, Klayto José Gonçalves dos Santos, Lysa Bernardes Minasi, Vitória Alvarenga Nunes, Alex Silva da Cruz, Mariana Yukari Hayasaki Porsani.

**Visualization:** Danilo Conrado Silva, Aparecido Divino da Cruz, Erika Figueiredo Pereira, Lysa Bernardes Minasi.

**Writing – original draft:** Danilo Conrado Silva.

**Writing – review & editing:** Fabio Alves Teixeira, Erika Figueiredo Pereira, Lysa Bernardes Minasi, Mariana Yukari Hayasaki Porsani.

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
