## [Decision Letter · Decision Letter 0]

25 Aug 2025

Dear Dr. Conrado Silva,

We look forward to receiving your revised manuscript.

Kind regards,

Ewa Tomaszewska, DVM Ph.D

Academic Editor

PLOS ONE

Journal Requirements:

Reviewers' comments:

Reviewer's Responses to Questions

**Comments to the Author**

1. Is the manuscript technically sound, and do the data support the conclusions?

Reviewer #1: Yes

2. Has the statistical analysis been performed appropriately and rigorously?

Reviewer #1: Yes

3. Have the authors made all data underlying the findings in their manuscript fully available?

Reviewer #1: Yes

4. Is the manuscript presented in an intelligible fashion and written in standard English?

Reviewer #1: Yes

Reviewer #1: Dear Authors,

The experiment was well designed and conducted, and the manuscript is well documented and written. I would like to make a minor suggestion: since there was a statistical difference in the stratification of owners according to household income, it would be important to specify which currency was used in this stratification. I believe that the Brazilian currency may have been converted into US dollars; however, this was not made clear.

**Do you want your identity to be public for this peer review?** For information about this choice, including consent withdrawal, please see our Privacy Policy

Reviewer #1: **Yes: ** Luiz Roberto Biondi

---

## [Author Response · Author response to Decision Letter 1]

20 Oct 2025

Dear Editor and Reviewer,

We thank you for your careful evaluation of our manuscript and for the constructive feedback provided. We sincerely appreciate the opportunity to revise our work and respond to the comments raised during the review process.

Please find below our point-by-point response.

Reviewer #1

Comment:

“Since there was a statistical difference in the stratification of owners according to household income, it would be important to specify which currency was used in this stratification. I believe that the Brazilian currency may have been converted into US dollars; however, this was not made clear.”

Response:

Thank you for your helpful suggestion. We agree that it is important to clarify the currency used in the income stratification.

We have revised the manuscript to explicitly state that household income, originally reported in Brazilian reais (BRL), was converted into US dollars (USD) for reporting and analysis purposes. This information has been added to the “Materials and Methods” section under the subsection “Data Collection and Categorization of Cats and Owners”.

Change made in the manuscript (page 6, lines 129–131):

“For the purpose of stratifying owners by household income, amounts in the Brazilian currency, the Brazilian real, were converted into US dollars.”

We believe this amendment addresses the reviewer’s concern and improves the clarity of the manuscript.

We thank you again for your time and consideration. We hope our revised version meets the journal’s standards and look forward to your favorable response.

Sincerely,

Dr. Danilo Conrado Silva

on behalf of all co-authors

---

## [Decision Letter · Decision Letter 1]

10 Nov 2025

Household Survey on Prevalence and Risk Factors for Obesity in Owned Cats from Central Brazil

PONE-D-25-41354R1

Dear Dr. Danilo Conrado Silva,

We’re pleased to inform you that your manuscript has been judged scientifically suitable for publication and will be formally accepted for publication once it meets all outstanding technical requirements.

Kind regards,

Ewa Tomaszewska, DVM Ph.D

Academic Editor

PLOS ONE

Additional Editor Comments (optional):

Reviewers' comments:

Reviewer's Responses to Questions

**Comments to the Author**

Reviewer #1: All comments have been addressed

2. Is the manuscript technically sound, and do the data support the conclusions?

Reviewer #1: Yes

3. Has the statistical analysis been performed appropriately and rigorously?

Reviewer #1: Yes

4. Have the authors made all data underlying the findings in their manuscript fully available?

Reviewer #1: Yes

5. Is the manuscript presented in an intelligible fashion and written in standard English?

Reviewer #1: Yes

Reviewer #1: I have no additional comments on the article. The authors have accepted and incorporated my previous suggestions into the text.

**Do you want your identity to be public for this peer review?** For information about this choice, including consent withdrawal, please see our Privacy Policy

Reviewer #1: No

---

## [Editor Report · Acceptance letter]

PONE-D-25-41354R1

PLOS ONE

Dear Dr. Conrado Silva,

I'm pleased to inform you that your manuscript has been deemed suitable for publication in PLOS ONE. Congratulations! Your manuscript is now being handed over to our production team.

Kind regards,

on behalf of

Professor Ewa Tomaszewska

Academic Editor

PLOS ONE